# Status of Microbiota in Odontogenic Inflammatory Lesions and Dental Surgery Procedures Performed on an Outpatient Basis

**DOI:** 10.3390/antibiotics11081025

**Published:** 2022-07-30

**Authors:** Tadeusz Morawiec, Joanna Śmieszek-Wilczewska, Mateusz Bogacz, Magdalena Jędrusik-Pawłowska, Anna Bubiłek-Bogacz, Anna Mertas

**Affiliations:** 1Department of Dental Surgery, Faculty of Medical Sciences in Zabrze, Medical University of Silesia, Plac Akademicki 17, 41-902 Bytom, Poland; morawiec@comfortdent.pl (T.M.); mateusz.bogacz@gmail.com (M.B.); chirmag@wp.pl (M.J.-P.); bogacz.a@poczta.fm (A.B.-B.); 2Department of Microbiology and Immunology, Faculty of Medical Sciences in Zabrze, Medical University of Silesia in Katowice, Jordana 19, 41-808 Zabrze, Poland; amertas@sum.edu.pl

**Keywords:** microbiota, odontogenic inflammations, antibiotics

## Abstract

Inflammations of the facial part of the skull are most commonly caused by a bacterial infection. They are a frequently occurring pathological process, which results from a rich bacterial flora of the oral cavity, as well as diseased teeth and periodontal tissues. These inflammations have a primarily mixed character with the prevalence of anaerobic bacteria. Gangrene of the dental pulp is the most common odontogenic cause. In the case of inflammations of oral tissues an early and corrective treatment results in quick recovery. The purpose of this work was to assess the efficiency of empirical antibiotic therapy applied in patients with inflammations of oral tissues on the basis of a drug susceptibility profile of bacteria isolated from material extracted from inflammatory lesions. The research material consisted of smears collected from patients with existing acute inflammations in the oral cavity. The smear was collected from the bottom of the lesion after its prior surgical treatment and pus evacuation, and again, 7 days after surgery. In patients with acute odontogenic inflammations the recommended first-line therapy are extended-spectrum penicillins, characterized by a low risk of side effects and strong antimicrobial activity. In the study group, both clindamycin and amoxicillin exhibited high efficiency in treating acute odontogenic inflammatory lesions in the oral cavity.

## 1. Introduction

The term microbiome was first used by American geneticist and microbiologist Joshua Lederberg to describe the totality of microorganisms present in a given habitat, both symbiotic and pathogenic [1]. In the body of an average adult weighing 70 kg reside microorganisms with a total mass of about 200 g, this is nearly 3.8 × 10^13^ cells of bacteria and other microorganisms, without which the proper functioning of the human body would not be possible. The relationship between the microbiome and the host is dynamic [2]. The composition and properties of our microbiome can change under the influence of many aspects of modern life, such as diet, tobacco consumption and stress, causing a state in which this finely tuned ecosystem is no longer balanced [3].

Bacterial infections occurring in the head and neck area are extremely dangerous processes due to their location in an area of complex anatomy containing many vital organs. The oral cavity is a complex habitat for microorganisms, the type, distribution, and potential pathogenicity of which vary depending on the region where they reside. The area in which the infection develops is characterized by unfavorable anatomic conditions, i.e., proximity to adjacent topographic spaces, and lack of valves in the venous vessels. The above-mentioned factors promote a high dynamic of development of the infection, creating the risk of many life-threatening complications. It is very important to immediately implement treatment, primarily causal (removal of the tooth, initiating the root canal therapy) and/or symptomatic (incision and drainage of the purulent lesion). In case of accompanying systemic symptoms such as increasing facial swelling, trismus, significant enlargement and soreness of surrounding lymph nodes, difficulty swallowing, tachycardia (heart rate above 100 beats per minute), chills, increased body temperature, additional use of antibacterial and anti-inflammatory drugs is necessary [4,5,6]. Because of the above, some of the most commonly used drugs in oral surgery in addition to painkillers are antibiotics.

The literature data indicate that the microorganisms causing the type of infections described above include both aerobic and anaerobic, Gram-positive and Gram-negative bacteria.

## 2. Materials and Methods

Fifty-two patients were included in the study, among whom two equinumerous groups were identified:

Study group (26 patients): patients with acute oral inflammation in whom the antibiotic was ordered for therapeutic indications. These were patients diagnosed with:-Submucosal abscess requiring removal of the causative tooth as well as soft tissue incision and seton use, 15 patients, including 2 females and 13 males;-Periapical abscess requiring only removal of the causative tooth (drainage of the purulent content was obtained through the alveolus), 11 patients, including 3 females and 8 males.

Control group (26 patients): patients with no signs or symptoms of acute inflammation who presented to the Oral Surgery Clinic for elective surgery, removal of an impacted third molar.

Inclusion criteria for the study was the occurrence of acute inflammation in the oral cavity and perioral tissues, requiring surgical intervention with clear indications for antibiotic therapy. The exclusion criteria from the study were as follows: patients undergoing diagnosis and treatment of head and neck oncologic diseases, patients with severe systemic diseases that make surgical treatment impossible in an outpatient setting, patients with diffuse inflammatory infiltrates requiring hospitalization, pregnant women and patients undergoing antibiotic therapy within 3 months prior to surgery.

The study took into account the general health of the patients, the medications they were taking, use of substances, and the previous treatment (or lack thereof) of the inflammation that had occurred.

The material for microbiological studies consisted of swabs from the lesion area taken from patients in the study and control groups before introduction of pharmacological therapy. The material was collected at the Oral Surgery Clinic and General Dental Clinic in Bytom. Culture, identification, and evaluation of drug susceptibility of microorganisms were performed in Microbiological Laboratory in Department of Microbiology and Immunology in Zabrze, Medical University of Silesia in Katowice. The time from material collection to delivery to the laboratory was less than 2 h.

Microbiological studies were conducted using conventional methodology used in microbiological diagnostics. The material obtained from the patients was inoculated into aerobic and anaerobic bacterial culture media for multiplication and isolation of pure microbial colonies. Columbia agar solid medium supplemented with 5% sheep blood at 37 °C was used to grow aerobic bacteria at 37 °C. Anaerobic bacteria were grown on Schaedler K3 solid medium supplemented with 5% sheep blood at 37 °C under anaerobic conditions obtained using Genbaganaer kits (bioMérieux, Marcy l’Etoile, France). Following the isolation and multiplication of the cultured bacterial strains, their species identification was performed using reagent kits: ENTEROtest 24 N, NEFERMtest 24 N, STREPTOtest 24, STAPHYtest 24, ANAEROtest 23, OXItest, PYRAtest, and the computer software TNW_lite 6.5 for identification of microorganism species (Erba-Lachema, Brno, Czech Republic). The bioMérieux tests: Katalaza, Slidex Staph Kit, and API Candida (Marcy l’Etoile, France) were also used. The reading and analysis of the obtained results were performed according to the recommendations of the manufacturers of the diagnostic reagent kits. The drug susceptibility of the isolated bacterial strains was determined by the disk-diffusion method and by Etest strips (bioMérieux, Marcy l’Etoile, France). This step of the study was performed according to current European Committee on Antibiotic Susceptibility Testing (EUCAST) recommendations.

During the analysis of microorganisms isolated in the material collected from patients for drug susceptibility assessment, the species most likely to cause the infection (in the study group) or species that can be a potential cause of infection development after the surgical procedure (in the control group) were selected. These were identified as potentially pathogenic strains (SPP) and an antibiogram was prepared against them. When at least one strain from a species was identified as a potential cause of infection, that species was included in the potentially pathogenic group (GPP).

All patients underwent clinical and radiographic examinations consistent with current standards of practice. During the examination, a dental diagram was filled, based on the caries intensity index decayed, missing and filled teeth (DMFT) and treatment index F1 were calculated for each patient. Each patient was informed in detail about the type, procedures, and purpose of the study.

Before the procedure, patients in both groups underwent radiological examination in the form of periapical X-ray, panoramic X-ray, and volumetric tomography when warranted.

The study was conducted with prior approval from the Bioethics Committee of the Silesian Medical Chamber in Katowice, Poland, Resolution No. 45/2015of 21 December.

### 2.1. Treatment Procedure—Study Group

In patients in the study group who were found to have a submucosal, subperiosteal, or subcutaneous abscess, the first step of treatment was to incise the emerging purulent lesion to evacuate its contents with subsequent restorative or surgical treatment (root canal therapy or tooth extraction). In the case of periapical abscesses, the surgical stage of the treatment consisted only of removing the causative tooth. The next step in the procedure was to take a swab for microbiological examination, which was purulent content coming from inflammatory foci in the oral cavity. Immediately before obtaining the material, the area around the lesion was isolated with sterile gauze and then dried and disinfected. The first portion of pus was removed, and then a swab was taken from as deep as possible in the lesion using a sterile swab stick (Test I). In the absence of indications for incision of the lesion, when the discharge of purulent content was obtained from the alveolus after tooth extraction, the swab was taken from the alveolus (Figure 1 and Figure 2a–d). In each case the swab sticks were placed in transport medium for aerobic and anaerobic microorganisms.

All the patients were required to attend daily routine follow-up appointments over the next few days after surgery to assess proper healing of the lesion. Each appointment included the assessment of patient’s general and local condition and seton replacement. This type of lesion usually takes 7–10 days to heal. At the last visit, a follow-up swab was taken from the lesion area (Test II; Figure 2e,f).

In cases of oral inflammation accompanied by potentially life-threatening symptoms (rapidly increasing facial swelling, trismus, significant enlargement and soreness of the surrounding lymph nodes, and difficulty swallowing and breathing) or systemic symptoms such as tachycardia (with heart rate above 100 beats per minute), chills, or increased body temperature, empiric antibiotic therapy was ordered to supplement the patient’s treatment.

### 2.2. Treatment Procedure—Control Group

The control group consisted of patients presenting for elective removal of an impacted third molar. The first step in the procedure was to provide oral hygiene instruction, assess hygiene indexes, and provide patients with preoperative recommendations. On the day of the scheduled surgery, a clinical and radiological examination (described above) was performed, and then a swab was taken for microbiological studies from the operated area, Test I. The next step was to perform surgery according to current guidelines. A follow-up visit was scheduled on day 7 after surgery. After evaluation of the patient’s clinical status, a swab was taken from the operated area, Test II. At the end of the visit, sutures were removed under topical anesthesia. If the procedure required an infringement of bone continuity or lasted longer than 30 min, empirical antibiotic therapy was ordered postoperatively.

In both groups of patients, when antibiotic therapy was indicated, the drugs ordered were amoxicillin with clavulanic acid or clindamycin.

### 2.3. Statistical Analysis

The statistical analysis included quantitative continuous (age, DMFT, F1, and API), discrete (microbial count), and qualitative variables (group type, antibiotic, cause, and healing). Statistical evaluation of the relationship between qualitative variables was performed using the Chi2 test of independence or the Chi2 test with Yates’ correction when there was at least one expected number less than five in the four-field table. For four-field tables with total counts less than 40, Fisher’s exact test was used. Quantitative variables were assessed for normal distribution using the Shapiro–Wilk test. Recognizing the normal distribution or at least symmetry of distribution allowed us to assess the significance of differences between the arithmetic means for two independent samples by Student’s *t*-test, while for the paired variables the dependent samples *t*-test was used. Homogeneity of variance was checked by Levene’s test. In the case of discrete variables two independent samples were compared using the Mann–Whitney U test, while two dependent samples were compared using the Wilcoxon pairwise rank order test. Descriptive characteristics of quantitative continuous variables were presented in the form of arithmetic mean and standard deviation (SD), while for quantitative discrete variables median, extreme values (minimum and maximum) and quartiles (lower and upper) were provided. Statistical analyses were performed using the Statistica PL v. 13 software (Statsoft, Krakow, Poland) assuming the significance level of α = 0.05.

## 3. Results and Discussion

### 3.1. Characteristic of Examined Population

The group of patients with acute odontogenic inflammation (study group) consisted of 26 patients including 5 females and 21 males, aged between 21 and 82 years (47.46 ± 14.49). The group of patients with a healthy mouth who presented for elective surgery (control group) consisted of 26 patients including 19 females and 7 males, aged between 13 and 82 years (33.04 ± 16.75). The mean age of patients in the study group was higher than in the control group, the difference was statistically significant (*p* = 0.0017). Patient age data are summarized in Table 1.

### 3.2. Microbiological Test Results

For analysis of the results obtained, the cultured bacteria were divided into four groups:-Gram-positive aerobic bacteria;-Gram-positive anaerobic bacteria;-Gram-negative aerobic bacteria;-Gram-negative anaerobic bacteria.

All cultured bacterial strains belonging to one of the above groups were subjected to comparative analysis for each test (Test I vs. Test II) and for each patient group (control group vs. study group), each time reporting the percentage of cultured bacteria of a particular group relative to all cultured bacteria. Due to significant differences in the number of cultured strains between species, in order to accurately assess the diversity of cultured bacterial flora, the above analysis was repeated considering only the number of cultured bacterial species for each of the four groups.

A total of 397 bacterial strains were cultured from material collected from the patients, 190 in the study group and 207 in the control group. The cultured strains belonged to 87 species, of which 64 species were found in the study group and 53 in the control group. In both the study and control groups, microorganisms from the flora physiologically present in the oral cavity were the most frequent: *Neisseria subflava* (82 strains), *Streptococcus mitis* (51 strains), *Streptococcus salivarius* (31 strains), and *Streptococcus sanguinis* (28 strains). In both groups of patients, in both Test I and Test II, aerobic Gram-positive bacterial strains were the most commonly cultured (the majority of bacterial species identified also belonged to this group). In Test II, there was a reduction in the total number of bacterial strains cultivated, as well as the number of species isolated, in both the study and control groups compared with Test I.

To determine which group of microorganisms was most often responsible for the development of infection, the number of cultured SPPs was compared with all cultured microorganisms of a given group. The results were evaluated against each test (Test I vs. Test II) and patient group (study group vs. control group). As before, the number of cultured GPPs was also compared with all cultured bacterial species. 

Among SPP in both patient groups, *Actinomyces naeslundii* was the most common species (17 strains). Moreover, in the study group *Propionibacterium propionicum* (4 strains) was among the most frequently isolated species, while in the control group they included *Actinomyces odontolyticus* (4 strains), and *Klebsiella oxytoca* (3 strains). 

In the study group, anaerobic Gram-positive bacteria comprised the highest percentage of SPP among all cultured strains. The lowest species diversity was observed for aerobic Gram-negative bacteria; however, as many as 75.00% (in Test I) and 66.67% (in Test II) of them belonged to GPP. Aerobic Gram-positive bacteria had the lowest percentage of SPPs (2.50% in Test I and 5.13% in Test II), and the only GPPs found were: *Streptococcus pneumoniae* and *Staphylococcus epidermidis MSCNS.*

Moreover, in the control group anaerobic Gram-positive bacteria had the highest percentage of SPP among all cultured strains. There was an increase in the number of cultured *Actinomyces naeslundii* strains between Test I (2 strains) and Test II (7 strains). Similar to the study group, a significant amount of GPP was found here among aerobic Gram-negative bacteria (40.00% in Test I and 83.33% in Test II). The only GPPs among the aerobic Gram-positive bacteria in this group were: *Streptococcus pneumoniae* and *Staphylococcus aureus MSSA.*


Analyzing the mean number of SPP of all the mentioned groups of bacteria between Test I and II, it was found to increase in both the study group (17.53% vs. 21.51%) and the control group (12.84% vs. 16.33%). The greatest increase in SPPs was observed in anaerobic Gram-positive bacteria. The greatest reduction in SPPs was found in anaerobic Gram-negative bacteria, no SPPs were cultured among this group of bacteria in Test II.

All results obtained for the isolated microbiota are presented by strain and SPP (Table 2 and Table 3) and by species and GPP (Table 4 and Table 5).

Comparing all bacterial strains cultured in both patient groups in Test I and II, the highest percentage of SPP was found for Gram-positive anaerobic bacteria (51.85%). The lowest percentage of SPPs was cultured among Gram-negative anaerobic (4.08%) and Gram-positive aerobic bacterial strains (4.82%). For aerobic bacteria, a higher percentage of SPP was found among Gram-negative bacteria compared with Gram-positive bacteria. The opposite was true for anaerobic bacteria, where a higher percentage of SPPs was found among Gram-positive bacteria. The differences found were statistically significant (Table 6).

Comparing all bacterial species cultured in both patient groups in Test I and II the highest percentage of GPP was found for Gram-negative aerobic bacteria. *Neisseria* spp. were the only species in this group that were not evaluated as a potential cause of inflammation. The lowest percentage of GPP was found among aerobic Gram-positive bacteria, among which only *Streptococcus pneumoniae, Staphylococcus aureus,* and *Staphylococcus epidermidis* were identified as potential causes of inflammation. For aerobic bacteria, a significantly higher percentage of GPP was found among Gram-negative species, whereas for anaerobic bacteria, a higher percentage of GPP was found among Gram-positive bacteria. The differences found were statistically significant (Table 7).

Table 8, Table 9, Table 10 and Table 11 detail all cultured bacterial strains by test (Test I vs. Test II) and patient group (study group vs. control group). GPPs are highlighted in red.

The cultured *Candida albicans* strains are shown in Table 12.

## 4. Discussion

The bacterial flora of the oral cavity, with its abundance and diversity, is second only to that of the gastrointestinal tract [3]. It consists of more than 700 species of bacteria colonizing, among others, the surfaces of teeth and mucous membranes [1]. The knowledge of which groups the most common potentially pathogenic bacteria belong to allows the use of the most effective antibiotic possible; as in the case of odontogenic inflammations antibiotic therapy is most often implemented empirically [7]. Studies within last 5 years suggest that odontogenic inflammatory lesions are polymicrobial, which is confirmed by the results of our research [8,9,10].

With such a diverse bacterial flora, any immune dysfunction or other factors that create a risk of microbial penetration deep into the tissues promote the development of infections. In terms of rapidly increasing bacterial resistance to antibiotics, infections are now one of the most serious challenges in healthcare. There are many controversies over the treatment of odontogenic inflammatory lesions regarding the surgical procedure itself [11] as well as the issue of antibiotics abuse [12,13,14]. The above leads to the formation of multidrug resistant strains of bacteria which complicates the treatment of bacterial infections in general [15]. Despite the increasing public awareness of the prevention and prophylaxis of diseases of teeth, periodontal tissues and oral mucosa, as well as constantly improved methods of treatment and development of antibiotic therapy, inflammation of oral tissues is a common ailment observed in patients. These infections are most often mixed in nature with a predominance of anaerobic bacteria. Sousa et al. [16] examined the bacterial flora in the root canals of teeth that developed periapical abscess. Anaerobic bacteria (detected in 90% of root canals) and Gram-positive bacteria (in 98.3% of root canals) were predominantly found. The most common strains isolated from the root canals were *Anaerococcus prevotii* (37.0%), *Parvimonas micra* (32.0%), and *Fusobacterium necrophorum* (32.0%). The bacteria found in the root canals were found to be 100% sensitive to amoxicillin with clavulanic acid and clindamycin. Among patients with periapical or submucosal abscess, only one case of *Anaerococcus prevotii* was cultured in the study (patient with periapical abscess, Test I after extraction of causative tooth 35); the other microorganisms mentioned above were not isolated. Similar to the study by Sousa et al. [16], among the potentially pathogenic strains in the present study, there was also a significant predominance of anaerobic Gram-positive bacteria; however, aerobic Gram-positive bacteria predominated among all the strains cultured [16]. Łysakowska et al. [17] took microbiological swabs of 37 root canals from 33 patients with indications for root canal therapy. *Streptococcus* spp., *Enterococcus faecalis,* and *Propionibacterium acnes* were the most commonly isolated microorganisms. Secondary lesions requiring revision of endodontic treatment had a greater diversity of bacterial flora than primary lesions [17]. No *Enterococcus faecalis* or *Propionibacterium acnes* species were cultured in this study; however, *Streptococcus* spp. were the most commonly isolated bacteria. This highlights how many different ecological niches within the oral cavity are colonized by bacteria of this genus. Loyola-Rodriguez et al. [18] investigated the microbiota and its drug susceptibility in 60 children who presented with acute odontogenic inflammation. Swabs were taken from the root canals of deciduous teeth. The most common bacteria isolated were: *Streptococcus oralis* and *Prevotella intermedia* (75.0%); *Treponema denticola* and *Porphyromonas gingivalis* (48.3%); *Streptococcus mutans* (45.0%); and *Campylobacter rectus* and *Streptococcus salivarius* (40.0%). The study found a significant percentage of clindamycin-resistant strains (85.9% resistant strains), compared with amoxicillin (43.7% resistant strains) and amoxicillin with clavulanic acid (12% resistant strains) [18]. Götz et al. [19] investigated the microbiota and its drug susceptibility in purulent facial soft tissue infections requiring extra- or intraoral incision. As in the present study, the cultured microbiota consisted predominantly of the Gram-positive bacteria and viridans group streptococci (*Streptococcus viridans*) that constitute the natural oral microbiota. The shifts in the composition of subgingival communities are the major factor of development of periodontitis [8]. As proved by Lamonte et al. [20], subgingival microbiota are measurably elevated several years prior to progression of alveolar bone loss. Deterioration of the alveolar socket increases the risk of tooth loss and limits the treatment options of dentoalveolar fractures [21].

Bacterial infections of the oral tissues are conditions where immediate and proper management results in rapid recovery of the patient. It is particularly important that the correct treatment (decompression of the abscess) and antibiotic therapy be administered as early as possible in high doses [4]. In addition, the issue of induction of drug resistance and the associated decrease in efficacy of antibiotics in the general population has been increasingly addressed in the literature. The authors also point out the problem of imbalance of gastrointestinal flora (dysbiosis). According to Othoniel et al. [22], the administration of amoxicillin (500 mg every 8 h for 5 days) induces significant changes in the intestinal bacterial flora, and it takes about two months to return to its normal state. During this time, the body’s immune homeostasis is disrupted, which, combined with the selection of amoxicillin-resistant strains, makes the body susceptible to developing infectious diseases. This can cause delayed local infections that develop 16–79 days after tooth extraction. Bacteria of the genera *Fusobacterium* and *Prevotella* associated with this type of infection are often characterized by resistance to amoxicillin [22,23]. As proven, the presence of *Fusobacterium nucleatum* may be a factor accelerating the development of colorectal cancer [24,25,26,27].

The status of microbiota in the oral cavity may vary depending on the oral care products used, and the following can have a significant effect on the development of inflammatory condition [28,29]: Knowledge of microbiota status in odontogenic inflammatory lesions improves the quality of antibiotic treatment which is most often implemented empirically. Amoxicillin with clavulanic acid and clindamycin show the greatest efficacy in the treatment of odontogenic inflammations; however, compared with the 1999 studies, the percentage of strains sensitive to these drugs has reduced [30].

## 5. Conclusions

There were no statistically significant differences between the test and control groups in the number of bacterial strains cultured in either Test I or Test II. In each case, the highest number of cultured strains belonged to aerobic Gram-positive bacteria and the lowest to anaerobic Gram-negative bacteria;There were also no statistically significant differences between the test and control groups in the number of bacterial species cultured in either Test I or Test II. Gram-positive aerobic bacteria have the highest species diversity and Gram-negative aerobic bacteria the lowest;The highest percentage of potentially pathogenic strains was found for Gram-positive anaerobic bacteria and the lowest for Gram-positive aerobic bacteria;The highest percentage of potentially pathogenic species was found for Gram-negative aerobic bacteria and the lowest for Gram-positive aerobic bacteria. In the group of Gram-negative aerobic microorganisms, the only non-pathogenic bacteria were strains of *Neisseria* spp. The only pathogenic species found in the Gram-positive aerobic microbial group were *Staphylococcus aureus, Staphylococcus epidermidis*, and *Streptococcus pneumoniae*.

## Figures and Tables

**Figure 1 antibiotics-11-01025-f001:**
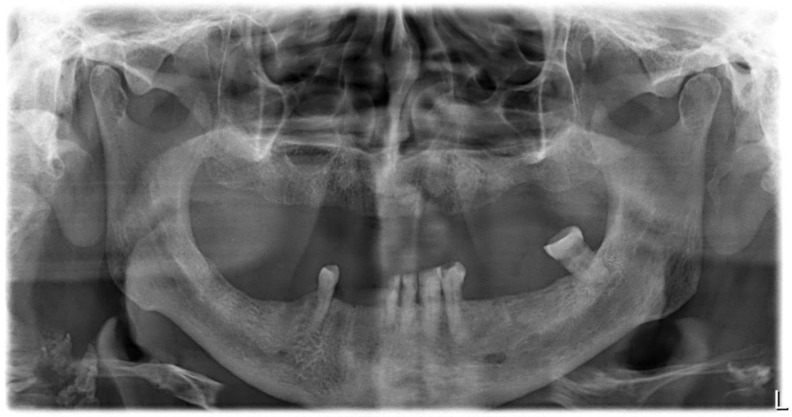
Panoramic X-ray showing bone translucency near the root apices of tooth 38, most likely a radiographic sign of exacerbated periapical tissue inflammation around tooth 38.

**Figure 2 antibiotics-11-01025-f002:**
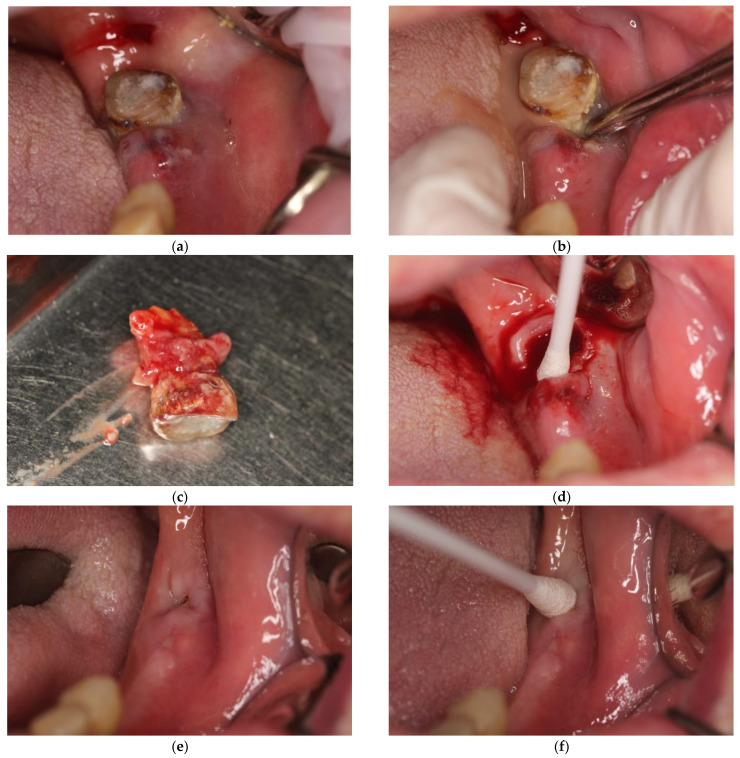
(**a**) Patient presented in (**e**), intraoral view. Visible submucosal abscess, forming near tooth 38. (**b**) This picture was taken after an attempt to luxate tooth 38 with a Bein root lever, resulting in the release of purulent contents. (**c**) Extracted tooth 38. Visible granulomatous and cystic lesions covering the tooth root. (**d**) situation after removal of tooth 38, resulting in complete emptying of the pus reservoir. No incision was required, and swab was taken from deep within the alveolus. (**e**) Patient’s clinical status at follow-up visit 7 days after extraction of tooth 38. (**f**) Taking a swab for bacteriological examination from the area after tooth extraction of tooth 38.

**Table 1 antibiotics-11-01025-t001:** Age of patients in the study and control groups.

	Study Group (Mean ± SD)	Control Group (Mean ± SD)	*p*
Age of patients (years)	47.5 ± 14.5	33.0 ± 16.8	0.0017

**Table 2 antibiotics-11-01025-t002:** Percentage of cultured bacterial strains from a given group relative to all cultured strains in a given test (Test I vs. Test II), in a given patient group (study vs. control).

Bacterial Strains		Study Group % (N_bact_/∑_bact_)	Control Group % (N_bact_/∑_bact_)	*p*
Anaerobic Gram-positive	Test I	23.71% (23/97)	17.43% (19/109)	0.2641
	Test II	23.66% (22/93)	17.35% (17/98)	0.2796
	*p*	0.9928	0.9873	–
Anaerobic Gram-negative	Test I	10.31% (10/97)	16.51% (18/109)	0.0787
	Test II	9.68% (9/93)	12.24% (12/98)	0.5708
	*p*	0.7915	0.3837	–
Aerobic Gram-positive	Test I	41.24% (40/97)	42.20% (46/109)	0.8885
	Test II	41.94% (39/93)	41.84% (41/98)	0.9890
	*p*	0.9222	0.9576	–
Aerobic Gram-negative	Test I	24.74% (24/97)	23.85% (26/109)	0.8819
	Test II	24.73 (23/93)	28.57% (28/98)	0.5488
	*p*	0.9986	0.4402	–

N_bact_—number of bacterial strains of a given group cultured in a given test and in a given group of patients; ∑_bact_—number of all bacterial strains cultured in a given test and in a given patient group.

**Table 3 antibiotics-11-01025-t003:** Percentage of cultured SPPs relative to all bacterial strains of a given group, cultured in a given test and in a given patient group.

SPP		Study Group % (N_SPP_/N_bact_)	Control Group % (N_SPP_/N_bact_)	*p*
Anaerobic Gram-positive	Test I	47.83% (11/23)	42.11% (8/19)	0.7141
	Test II	59.09% (13/22)	58.82% (10/17)	0.7555
	*p*	0.4490	0.5043	–
Anaerobic Gram-negative	Test I	10.00% (1/10)	12.50% (1/18)	0.5952
	Test II	0.00% (0/9)	0.00% (0/12)	1.0000
	*p*	0.3704	0.6000	–
Aerobic Gram-positive	Test I	2.50% (1/40)	4.35% (2/46)	0.9019
	Test II	5.13% (2/39)	2.44% (1/41)	0.9648
	*p*	0.9822	0.9192	–
Aerobic Gram-negative	Test I	16.67% (4/24)	11.54% (3/26)	0.9091
	Test II	17.39% (4/23)	17.86% (5/28)	0.7447
	*p*	0.7474	0.7874	–

N_bact_—number of bacterial strains of a given group cultured in a given test and in a given group of patients; N_SPP_—number of bacterial SPPs of a given group cultured in a given test and in a given patient group.

**Table 4 antibiotics-11-01025-t004:** Percentage of cultured bacterial species from a given group relative to all cultured species in a given test (Test I vs. Test II), in a given patient group (study vs. control).

Bacterial Species		Study Group % (N_sp_/∑_sp_)	Control Group % (N_sp_/∑_sp_)	*p*
Anaerobic Gram-positive	Test I	35.00% (14/40)	28.95% (11/38)	0.5670
	Test II	34.21% (13/38)	25.71% (9/35)	0.4293
	*p*	0.9416	0.7386	–
Anaerobic Gram-negative	Test I	17.50% (7/40)	18.42% (7/38)	0.9162
	Test II	15.79% (6/38)	17.14% (6/35)	0.8770
	*p*	0.8405	0.8874	–
Aerobic Gram-positive	Test I	37.50% (15/40)	39.47% (15/38)	0.8579
	Test II	34.21% (13/38)	40.00% (14/35)	0.6087
	*p*	0.7621	0.9634	–
Aerobic Gram-negative	Test I	9.76% (4/40)	13.16% (5/38)	0.9348
	Test II	15.79% (6/38)	17.14% (6/35)	0.8770
	*p*	0.6704	0.6368	–

N_sp_—number of bacterial species of a given group cultured in a given test and in a given group of patients; ∑sp—number of all bacterial species cultured in a given test and in a given patient group.

**Table 5 antibiotics-11-01025-t005:** Percentage of cultured GPPs relative to all bacterial species of a given group, cultured in a given test and in a given patient group.

GPP		Study Group % (N_GPP_/N_sp_)	Control Group % (N_GPP_/N_sp_)	*p*
Anaerobic Gram-positive	Test I	42.86% (6/14)	54.55% (6/11)	0.8592
	Test II	53.85% (7/13)	33.33% (3/9)	0.3050
	*p*	0.8528	0.3110	–
Anaerobic Gram-negative	Test I	14.29% (1/7)	14.29% (1/7)	0.7692
	Test II	0.00% (0/6)	0.00% (0/6)	1.0000
	*p*	0.5385	0.5385	–
Aerobic Gram-positive	Test I	6.67% (1/15)	13.33% (2/15)	0.5000
	Test II	15.38% (2/13)	7.14% (1/14)	0.4711
	*p*	0.4444	0.5268	–
Aerobic Gram-negative	Test I	75.00% (3/4)	40.00% (2/5)	0.3571
	Test II	66.67% (4/6)	83.33% (5/6)	0.5000
	*p*	0.6667	0.1970	–

N_sp_—number of bacterial species of a given group cultured in a given test and in a given group of patients; N_GPP_—number of bacterial GPPs of a given group cultured in a given test and in a given patient group.

**Table 6 antibiotics-11-01025-t006:** Percentage of SPP relative to all bacterial strains cultured in this study.

	Aerobic	Anaerobic	*p*
Gram-positive	4.82% (8/166)	51.85% (42/81)	<0.0001
Gram-negative	15.84% (16/101)	4.08% (2/49)	0.0383
*p*	0.0023	<0.0001	–

**Table 7 antibiotics-11-01025-t007:** Percentage of GPP relative to all bacterial species cultured in this study.

	Aerobic	Anaerobic	*p*
Gram-positive	10.00% (3/30)	48.15% (13/27)	0.0015
Gram-negative	73.33% (11/15)	15.38% (2/13)	0.0072
*p*	<0.0001	0.0458	–

**Table 8 antibiotics-11-01025-t008:** Number of Gram-positive, anaerobic microbial strains. GPPs are highlighted in red.

Anaerobic Gram-Positive						
Bacterial Species	Study Group			Control Group		
	Test I	Test II	Total	Test I	Test II	Total
*Actinomyces naeslundii*	4	4	8	2	7	9
*Bifidobacterium odolescentis*	0	4	4	3	1	4
*Atopobium minutum*	3	1	4	0	1	1
*Propionibacterium propionicum*	0	4	4	0	0	0
*Bifidobacterium dentium*	2	1	3	3	2	5
*Clostridium perfringens*	1	1	2	1	0	1
*Actinomyces israelii*	2	0	2	1	0	1
*Peptococcus niger*	2	0	2	0	0	0
*Actinomyces meyeri*	2	0	2	0	0	0
*Bifidobacterium breve*	1	0	1	3	1	4
*Clostridium sporogenes*	0	1	1	0	1	1
*Faclamia sourekii*	0	1	1	0	1	1
*Clostridium novyi biovar A*	1	0	1	1	0	1
*Abiotrophia odicens*	1	0	1	0	0	0
*Anaerococcus prevotii*	0	1	1	0	0	0
*Clostridium butyricum*	0	1	1	0	0	0
*Clostridium chauvoei*	0	1	1	0	0	0
*Clostridium tertium*	1	0	1	0	0	0
*Pseudoramibacter alactolyticus*	1	0	1	0	0	0
*Actinomyces viscosus*	0	1	1	0	0	0
*Lactobacillus acidophilus*	0	1	1	0	0	0
*Lactobacillus catenaformis*	1	0	1	0	0	0
*Lactobacillus jensenii*	1	0	1	0	0	0
*Actinomyces odontolyticus*	0	0	0	2	2	4
*Bifidobacterium longum*	0	0	0	1	1	2
*Clostridium novyi*	0	0	0	1	0	1
*Eubacterium limosum*	0	0	0	1	0	1
**Total**	**23**	**22**	**45**	**19**	**17**	**36**

**Table 9 antibiotics-11-01025-t009:** Number of Gram-negative, anaerobic microbial strains. GPPs are highlighted in red.

Anaerobic Gram-Negative						
Bacterial Species	Study Group			Control Group		
	Test I	Test II	Total	Test I	Test II	Total
*Campylobacter gracilis*	1	3	4	3	2	5
*Campylobacter ureolyticus*	2	2	4	1	1	2
*Pseudoflavonifractor capillosus*	3	0	3	0	1	1
*Veillonella parvula*	1	1	2	4	0	4
*Mitsuokella multacida*	1	0	1	6	5	11
*Prevotella oralis*	1	0	1	0	1	1
*Fusobacterium mortiferum*	0	1	1	0	0	0
*Fusobacterium nucleatum*	1	0	1	0	0	0
*Prevotella intermedia*	0	1	1	0	0	0
*Parabacteroides distasonis*	0	1	1	0	0	0
*Capnocytophaga ochracea*	0	0	0	2	2	4
*Bacteroides ovatus*	0	0	0	1	0	1
*Prevotella melaninogenica*	0	0	0	1	0	1
**Total**	**10**	**9**	**19**	**18**	**12**	**30**

**Table 10 antibiotics-11-01025-t010:** Number of Gram-positive, aerobic microbial strains. GPPs are highlighted in red.

Aerobic Gram-Positive						
Bacterial Species	Study Group			Control Group		
	Test I	Test II	Total	Test I	Test II	Total
*Streptococcus mitis*	12	14	26	12	13	25
*Streptococcus sanguinis*	7	7	14	8	6	14
*Streptococcus salivarius*	5	6	11	12	8	20
*Sarcina spp.*	2	1	3	0	1	1
*Staphylococcus epidermidis MSCNS*	2	1	3	0	0	0
*Streptococcus acidominimus*	2	0	2	1	2	3
*Staphylococcus lentus*	2	0	2	1	0	1
*Streptococcus anginosus*	0	2	2	0	1	1
*Streptococcus pneumoniae*	0	2	2	1	0	1
*Streptococcus suis*	1	1	2	0	0	0
*Streptococcus intermedius*	1	0	1	2	1	3
*Streptococcus mutans*	1	0	1	2	0	2
*Aerococcus viridans*	1	0	1	0	1	1
*Enterococcus malodoratus*	0	1	1	1	0	1
*Gemella spp.*	1	0	1	0	1	1
*Enterococcus columbae*	1	0	1	0	0	0
*Enterococcus hirae*	0	1	1	0	0	0
*Staphylococcus caprae*	0	1	1	0	0	0
*Staphylococcus chromogenes*	1	0	1	0	0	0
*Staphylococcus hominis*	1	0	1	0	0	0
*Staphylococcus sciuri*	0	1	1	0	0	0
*Staphylococcus warneri*	0	1	1	0	0	0
*Streptococcus uberis*	0	0	0	1	2	3
*Enterococcus gallinarum*	0	0	0	0	2	2
*Staphylococcus aureus MSSA*	0	0	0	1	1	2
*Streptococcus canis*	0	0	0	1	1	2
*Enterococcus casseliflavus*	0	0	0	0	1	1
*Globicatella sanquinis*	0	0	0	1	0	1
*Lactococcus lactis*	0	0	0	1	0	1
*Streptococcus bovis biovar I*	0	0	0	1	0	1
**Total**	**40**	**39**	**79**	**46**	**41**	**87**

**Table 11 antibiotics-11-01025-t011:** Number of Gram-negative, aerobic microbial strains. GPPs are highlighted in red.

Aerobic Gram-Negative						
Bacterial Species	Study Group			Control Group		
	Test I	Test II	Total	Test I	Test II	Total
*Neisseria subflava*	20	18	38	21	23	44
*Enterobacter cloacae*	2	0	2	0	0	0
*Escherichia coli*	1	1	2	0	0	0
*Klebsiella pneumoniae*	0	1	1	0	1	1
*Enterobacter kobeii*	0	1	1	0	0	0
*Hafnia alvei*	0	1	1	0	0	0
*Neisseria caprae*	0	1	1	0	0	0
*Serratia odorifera*	1	0	1	0	0	0
*Klebsiella oxytoca*	0	0	0	2	1	3
*Burkholderia cepacia*	0	0	0	0	1	1
*Chryseobacterium indologenes*	0	0	0	1	0	1
*Enterobacter aerogenes*	0	0	0	0	1	1
*Neisseria flavescens*	0	0	0	1	0	1
*Neisseria sicca*	0	0	0	1	0	1
*Providencia rustigianii*	0	0	0	0	1	1
**Total**	**24**	**23**	**47**	**26**	**28**	**54**

**Table 12 antibiotics-11-01025-t012:** Number of Candida albicans strains.

Bacterial Species	Study Group			Control Group		
	Test I	Test II	Total	Test I	Test II	Total
*Candida albicans*	5	5	**10**	4	3	**7**

## Data Availability

The data presented in this study are available from the corresponding authors.

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
