# Peer review of "Status of Microbiota in Odontogenic Inflammatory Lesions and Dental Surgery Procedures Performed on an Outpatient Basis"

_antibiotics, 2022, doi:10.3390/antibiotics11081025_

Round 1

Reviewer 1 Report

The work is well written, clear and concise for the reader.

My recommendation would be to review the bibliography and add a new reference that highlights the role of periodontal disease. Thank you.

Fernández-Ferro M, Fernández-Sanromán J, Costas-López A, López-Betancourt A, Casañas-Villalba N, López-Fernández P. Complex Dentoalveolar Fractures: Main Clinical Variables Description and Analysis. J Craniofac Surg. 2020 Nov/Dec;31(8):e761-e765. doi: 10.1097/SCS.0000000000006711. PMID: 33136904.

Author Response

Dear Editor,
Thank you for the detailed analysis and assessment of the article entitled „Status of microbiota in 
odontogenic inflammatory lesions and dental surgery procedures performed on an outpatient 
basis” authored by Tadeusz Morawiec, Joanna Åšmieszek-Wilczewska, Mateusz Bogacz, Magdalena 
JÄ™drusik-PawÅ‚owska, Anna BubiÅ‚ek-Bogacz and Anna Mertas. The following corrections and additions 
were made, as recommended:
1. The bibliography was updated and supplemented with the latest articles.
2. Sugested referrence regarding periodontal disease was added.
3. Sugested referrence regarding odontogenic inflammations was added.
4. Discussion was improved, more recent articles on a related subject are included.
5. Manuscript undergoes additional review by native English speaker.
6. Articles mentioned by the reviewer no. 2 are part of a larger research work, of which the earlier 
one presents the results of the pilot research. There may be similarities between the abovementioned works, although the authors have made every effort to avoid them.
7. Sugested modifications in materials and methods section were implemented.
8. Tables were re-formatted. The highest scores are now listed as first.
9. Photo 2d shows correct procedure of swab collecting. After tooth extraction, the pus was 
removed and the swab was taken from bloody exudate containing viable microorganisms and 
suitable for bacteriological examination.
10. Conclusions were improved.
11. Author contributions were described.
All corrections and additions were highlighted in blue.
I hereby kindly ask you to accept the corrected article for publication in the „Antibiotics” (MDPI).
Sincerely,
Joanna Åšmieszek-Wilczewska, PhD
Medical Uniwersity of Silesia in Katowice,
Department of Dental Surgery,
Pl. Akademicki 17,
41-902 Bytom,
Poland
Phone: +48606410639
E-mail: [email protected]

Reviewer 2 Report

Status of microbiota in odontogenic inflammatory lesions and dental surgery procedures performed on an outpatient basis 

By Tadeusz Morawiec 1 , Joanna Śmieszek-Wilczewska 1,* , Mateusz Bogacz 1 , Magdalena Jędrusik-Pawłowska 1 , Anna 4 Bubiłek-Bogacz 1 and Anna Mertas 2

Comments to the Author
 The topic is of interest, and the manuscript is well illustrated. However, the clarity of the manuscript should be substantially improved.

I mean (1) clarity in relation to the novelty of this article and (2) clarity of particular statements.

1. Are there controversies in this field? What are the most recent and important achievements in the field?  In my opinion, answers to these questions should be emphasized. Perhaps, in some cases, the novelty of the recent achievements should be highlighted by indicating the year of publication in the text of the manuscript.

2. The results and discussion section is very weak and no emphasis is given on the discussion of the results like why certain effects are coming into existence and what could be the possible reason behind them?

3. Results and conclusion. The section devoted to the explanation of the results suffers from the same problems revealed so far. Your storyline in the results section (and conclusion) is hard to follow. Moreover, the conclusions reached are really far from what one can infer from the empirical results. The discussion should be rather organized around arguments avoiding simply describing details without providing much meaning. A real discussion should also link the findings of the study to theory and/or literature.

4. Spacing, punctuation marks, grammar, and spelling errors should be reviewed thoroughly. I found so many typos throughout the manuscript.

English is modest. Therefore, the authors need to improve their writing style. In addition, the whole manuscript needs to be checked by native English speakers.

It will be worthwhile to include the most relevant articles like the following:

[Bacterial flora of odontogenic and non-odontogenic inflammations of the oro-facial region].

Szontágh E, Szöke I, Nagy E, Mari A.Fogorv Sz. 1999 Feb;92(2):45-50.

What are % similarities with published articles in the literature according to iThenticate particularly with the following 2 references:

1. M. Bogacz, T. Morawiec, J. Åšmieszek-Wilczewska, A. Mertas, K. Janowska-Bogacz, M. WrzoÅ‚, P. Kownacki, R. Rój, M. Pala, K. 423 Wiatrak, T. Piekarz, A. Kowalczewska, E. KopczyÅ„ska, A. BibuÅ‚ek-Bogacz, I. Niedzielska, “Evaluation of drug susceptibility of 424 microorganisms in ondotogenic oral cavity inflammations.” Pol.J.Environ.Stud. vol. 25, no. 6A, pp. 122-126, 2016. 425

2. M. Bogacz, T. Morawiec, J. Åšmieszek-Wilczewska, K. Janowska-Bogacz, A. BubiÅ‚ek-Bogacz, R. Rój, K. Pinocy, A. Mertas, 426 “Evaluation of drug susceptibility of microorganisms in odontogenic inflammations and dental surgery procedures performed 427 on an outpatient basis.” Biomed Res.Int., Vol. 2019, ID 2010453, pp. 1-12, 2019.

Author Response

(The authors gave the same response as above.)

Reviewer 3 Report

Dear colleagues!

Thank you for the opportunity to review your research, the relevance of which is high.

In materials and methods, write male / female instead of man / woman.

In terms of the number of participants, it is necessary to write about the criteria for inclusion and exclusion from the study, and also to clarify whether this number was sufficient to obtain reliable results.

In photo 2 d - do you think this photo is correct for demonstrating the material sampling stage?

Results. If possible, I recommend that you re-format the tables for pathogens, putting the highest scores first.

In conclusion, I want to note that the authors contribution is not indicated, and also point out the need to update the references in the bibliography, since some of them were published more than 10 years ago.

Author Response

(The authors gave the same response as above.)
